# Efficient Communication in Multi-Agent Reinforcement Learning via Variance Based Control

**Sai Qian Zhang**
Harvard University

**Qi Zhang**
Amazon Inc.

**Jieyu Lin**
University of Toronto

## Abstract

Multi-agent reinforcement learning (MARL) has recently received considerable attention due to its applicability to a wide range of real-world applications. However, achieving efficient communication among agents has always been an overarching problem in MARL. In this work, we propose Variance Based Control (VBC), a simple yet efficient technique to improve communication efficiency in MARL. By limiting the variance of the exchanged messages between agents during the training phase, the noisy component in the messages can be eliminated effectively, while the useful part can be preserved and utilized by the agents for better performance. Our evaluation using multiple MARL benchmarks indicates that our method achieves $2 - 10\times$ lower in communication overhead than state-of-the-art MARL algorithms, while allowing agents to achieve better overall performance.

## 1   Introduction

Many real-world applications (*e.g.*, autonomous driving [16], game playing [12] and robotics control [9]) today require reinforcement learning tasks to be carried out in multi-agent settings. In MARL, multiple agents interact with each other in a shared environment. Each agent only has access to partial observations of the environment, and needs to make local decisions based on partial observations as well as both direct and indirect interactions with the other agents. This complex interaction model has introduced numerous challenges for MARL. In particular, during the training phase, each agent may dynamically change its strategy, causing dynamics in the surrounding environment and instability in the training process. Worse still, each agent can easily overfit its strategy to the behaviours of other agents [11], which may seriously deteriorate the overall performance.

In the research literature, there have been three lines of research that try to mitigate the instability and inefficiency caused by decentralized execution. The most common approach is independent Q-learning (IQL) [20], which breaks down a multi-agent learning problem into multiple independent single-agent learning problems, thus allowing each agent to learn and act independently. Unfortunately, this approach does not account for instability caused by environment dynamics, and therefore often suffer from the problem of poor convergence. The second approach adopts the centralized training and decentralized execution [18] paradigm, where a joint action value function is learned during the training phase to better coordinate the agents' behaviours. During execution, each agent acts independently without direct communication. The third approach introduces communication among agents during execution [17, 3]. This approach allows each agent to dynamically adjusts its strategy based on its local observation along with the information received from the other agents. Nonetheless, it introduces additional communication overhead in terms of latency and bandwidth during execution, and its effectiveness is heavily dependent on the usefulness of the received information.

In this work, we leverage the advantages of both the second and third approaches. Specifically, we consider a fully cooperative scenario where multiple agents collaborate to achieve a common objective. The agents are trained in a centralized fashion within the multi-agent Q-learning framework, and are allowed to communicate with each other during execution. However, unlike previous work,

we make a few key observations. First, for many applications, it is often superfluous for an agent to wait for feedback from all surrounding agents before making an action decision. For instance, when the front camera on an autonomous vehicle detects an obstacle within the dangerous distance limit, it triggers the 'brake' signal 2 without waiting for the feedback from the other parts of the vehicle. Second, the feedback received from the other agents may not always provide useful information. For example, the navigation system of the autonomous vehicle should pay more attention to the messages sent by the perception system (*e.g.*, camera, radar), and less attention to the entertainment system inside the vehicle before taking its action. The full (i.e., all-to-all) communication pattern among the agents can lead to a significant communication overhead in terms of both bandwidth and latency, which limits its practicality and effectiveness in real applications with strict latency requirements and bandwidth constraints (e.g., real-time traffic signal control, autonomous driving, etc). In addition, as pointed out by Jiang *et al.* [7], an excessive amount of communication may introduce useless and even harmful information which can even impair the convergence of the learning process.

Motivated by these observations, we design a novel deep MARL architecture that can significantly improve inter-agent communication efficiency. Specifically, we introduce *Variance Based Control (VBC)*, a simple yet efficient approach to reduce the among of information transferred between agents. By inserting an extra loss term on the variance of the exchanged information, the meaningful part of the messages can be effectively extracted and utilized to benefit the training of each individual agent. Furthermore, unlike previous work, we do not require an extra decision module to dynamically adjust the communication pattern. This allows us to reduce the model complexity significantly. Instead, each agent first makes a preliminary decision based on its local information, and initiates communication only when its confidence level on this preliminary decision is low. Similarly, upon receiving the communication request, the agent replies to the request only when its message is informative. By only exchanging useful information among the agents, VBC not only improves agent performance, but also substantially reduces communication overhead during execution. Lastly, it can be theoretically shown that the resulting training algorithm provides guaranteed stability.

For evaluation, we test VBC on several MARL benchmarks, including StarCraft Multi-Agent Challenge [15], Cooperative Navigation (CN) [10] and Predator-prey (PP) [8]. For StarCraft Multi-Agent Challenge, VBC achieves $20\%$ higher winning rate and $2-10\times$ lower communication overhead on average compared with the other benchmark algorithms. For both CN and PP scenarios, VBC outperforms the existing algorithms and incurs much lower overhead than existing communication-enabled approaches. A video demo is available at [2] for a better illustration of the VBC performance. The code is available at https://github.com/saizhang0218/VBC.

## 2   Related Work

The simplest training method for MARL is to make each agent learn independently using Independent Q-Learning (IQL) [20]. Although IQL is successful in solving simple tasks such as Pong [19], it ignores the environment dynamics arose from the interactions among the agents. As a result, it suffers from the problem of poor convergence, making it difficult to handle advanced tasks.

Given the recent success on deep Q-learning [12], some recent studies explore the scheme of centralized training and decentralized execution. Sunehag *et al.* [18] propose Value Decomposition Network (VDN), a method that acquires the joint action value function by summing up all the action value functions of each agent. All the agents are trained as a whole by updating the joint action value functions iteratively. QMIX [14] sheds some light on VDN, and utilizes a neural network to represent the joint action value function as a function of the individual action value functions and the global state information. The authors of [10] extend the actor-critic methods to the multi-agent scenario. By performing centralized training and decentralized execution over the agents, the agents can better adapt to the changes in the environment and collaborate with each other. Foerster *et al.* [5] propose counterfactual multi-agent policy gradient (COMA), which employs a centralized critic function to estimate the action value function of the joint, and decentralized actor functions to make each agent execute independently. All the aforementioned methods assume no communication between the agents during the execution. As a result, many subsequent approaches, including ours, can be applied to improve the performance of these methods.

Learning the communication pattern for MARL is first proposed by Sukhbaatar *et. al.* [17]. The authors introduce *CommNet*, a framework that adopts continuous communication for fully cooperative

tasks. During the execution, each agent takes their internal states as well as the means of the internal states of the rest agents as the input to make decision on its action. The BiCNet [13] uses a bidirectional coordinated network to connect the agents. However, both schemes require all-to-all communication among the agents, which can cause a significant communication overhead and latency.

Several other proposals [3, 7, 8] use a selection module to dynamically adjust the communication pattern among the agents. In Differentiable Inter-Agent Learning (DIAL) [3], the messages produced by an agent are selectively sent to the neighboring agents through the discretize/regularise unit (DRU). By jointly training DRU with the agent network, the communication overhead can be efficiently reduced. Jiang *et. al.* [7] propose an attentional communication model that learns when the communication is required and how to aggregate the shared information. However, an agent can only talk to the agents within its observable range at each timestep. This limits the speed of information propagation, and restricts the possible communication patterns when the local observable field is small. Kim *et. al.* [8] propose a communication scheduling scheme for wireless environment, but only a fraction of the agents can broadcast their messages at each time. In comparison, our approach does not impose hard constraints on the communication pattern, which is beneficial to the learning process. Also our method does not adopt additional decision module for the communication scheduling, which greatly reduces the model complexity.

## 3   Background

**Deep Q-networks:**  We consider a standard reinforcement learning problem based on Markov Decision Process (MDP). At each timestamp $t$, the agent observes the state $s_t$, and chooses an action $a_t$. It then receives a reward $r_t$ for its action $a_t$ and proceeds to the next state $s_{t+1}$. The goal is to maximize the total expected discounted reward $R = \sum_{t=1}^{T} \gamma^t r_t$, where $\gamma \in [0, 1]$ is the discount factor. A Deep Q-Network (DQN) use a deep neural network to represent the action value function $Q_\theta(s, a) = E[R_t | s_t = s, a_t = a]$, where $\theta$ represents the parameters of the neural network, and $R_t$ is the total rewards received at and after $t$. During the training phase, a replay buffer is used to store the transition tuples $\langle s_t, a_t, s_{t+1}, r_t \rangle$. The action value function $Q_\theta(s, a)$ can be trained recursively by minimizing the loss $L = E_{s_t, a_t, r_t, s_{t+1}}[y_t - Q_\theta(s_t, a_t)]^2$, where $y_t = r_t + \gamma max_{a_{t+1}} Q_{\theta'}(s_t, a_{t+1})$ and $\theta'$ represents the parameters of the *target network*. An action is usually selected with $\epsilon$-greedy policy. Namely, selecting the action with maximum action value with probability $1 - \epsilon$, and choosing a random action with probability $\epsilon$.

**Multi-agent deep reinforcement learning:**  We consider an environment with $N$ agents work cooperatively to fulfill a given task. At timestep $t$, each agent $i$ ($1 \leq i \leq N$) receives a local observation $o_i^t$ and executes an action $a_i^t$. They then receive a joint reward $r_t$ and proceed to the next state. We use a vector $\mathbf{a}_t = \{a_i^t\}$ to represent the joint actions taken by all the agents. The agents aim to maximize the joint reward by choosing the best joint actions $\mathbf{a}_t$ at each timestep $t$.

**Deep recurrent Q-networks:** Traditional DQNs generate action solely based on a limited number of local observations without considering the prior knowledge. Hausknecht *et al.* [6] introduce Deep Recurrent Q-Networks (DRQN), which models the action value function with a recurrent neural network (RNN). The DRQN leverages its recurrent structure to integrate the previous observations and knowledge for better decision-making. At each timestep $t$, the DRQN $Q_\theta(o_i^t, h_i^{t-1}, a_i^t)$ takes the local observation $o_i^t$ and hidden state $h_i^{t-1}$ from the previous steps as input to yield action values.

**Learning the joint Q-function:**  Recent research effort has been made on the learning of joint action value function for multi-agent Q-learning. Two representative works are VDN [18] and QMIX [14]. In VDN, the joint action value function $Q_{tot}(\mathbf{o}_t, \mathbf{h}_{t-1}, \mathbf{a}_t)$ is assumed to be the sum of all the individual action value functions, i.e. $Q_{tot}(\mathbf{o}_t, \mathbf{h}_{t-1}, \mathbf{a}_t) = \sum_i Q_i(o_i^t, h_i^{t-1}, a_i^t)$, where $\mathbf{o}_t = \{o_i^t\}$, $\mathbf{h}_t = \{h_i^t\}$ and $\mathbf{a}_t = \{a_i^t\}$ are the collection of the observations, hidden states and actions of all the agents at timestep $t$ respectively. QMIX employs a neural network to represent the joint value function $Q_{tot}(\mathbf{o}_t, \mathbf{h}_{t-1}, \mathbf{a}_t)$ as a nonlinear function of $Q_i(o_i^t, h_i^{t-1}, a_i^t)$ and global state $s_t$.

## 4   Variance Based Control

In this section, we present the detailed design of VBC in the context of multi-agent Q-learning. The main idea of VBC is to improve agent performance and communication efficiency by limiting the

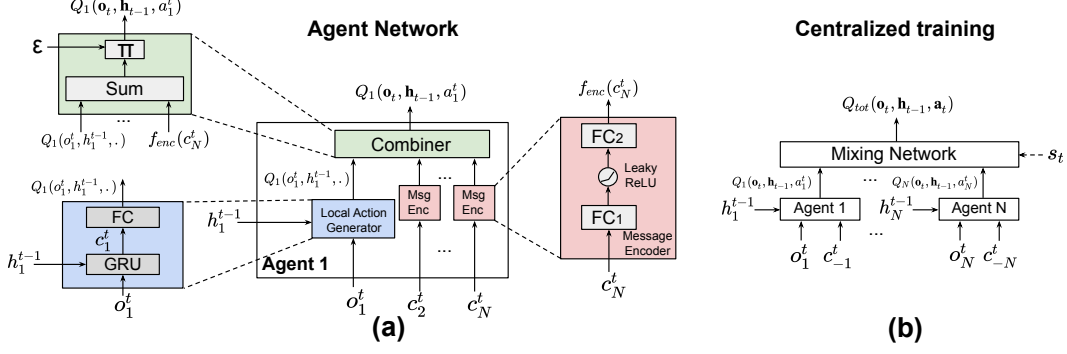

Figure 1: (a) Agent network structure of agent 1, which consists of local agent generator, combiner and several message encoder. (b) The mixing network takes the output $Q_i(\mathbf{o}_t, \mathbf{h}_{t-1}, a_i^t)$ from each network of agent $i$, and perform centralized training. $c_{-i}^t$ means all the $c_{j \neq i}^t$.

variance of the transferred messages. During execution, each agent communicates with other agents only when its local decision is ambiguous. The degree of ambiguity is measured by the difference between the top two largest action values. Upon receiving the communication request from other agents, the agent replies only if its feedback is informative, namely the variance of the feedback is high.

## 4.1 Agent Network Design

The agent network consists of the following three networks: local action generator, message encoder and combiner. Figure 1(a) describes the network architecture for agent 1. The *local action generator* consists of a Gated Recurrent Unit (GRU) and a fully connected layer (FC). For agent $i$, the GRU takes the local observation $o_i^t$ and the hidden state $h_i^{t-1}$ as the inputs, and generates the intermediate results $c_i^t$. $c_i^t$ is then sent to the FC layer, which outputs the local action values $Q_i(o_i^t, h_i^{t-1}, a_i^t)$ for each action $a_i^t \in A$, where $A$ is the set of possible actions. The *message encoder*, $f_{enc}^{ij}(.)$, is a multi-layer perceptron (MLP) which contains two FC layers and a leaky ReLU layer. The agent network involves multiple independent message encoders, each accepts $c_j^t$ from another agent $j$ ($j \neq i$), and outputs $f_{enc}^{ij}(c_j^t)$. The outputs from local action generator and message encoder are then sent to the *combiner*, which produces the global action value function $Q_i(\mathbf{o}_t, \mathbf{h}_{t-1}, a_i^t)$ of agent $i$ by taking into account the global observation $\mathbf{o}_t$ and global history $\mathbf{h}_{t-1}$. To simplify the design and reduce model complexity, we do not introduce extra parameters for the combiner. Instead, we make the dimension of the $f_{enc}^{ij}(c_j^t)$ the same as the local action values $Q_i(o_i^t, h_i^{t-1}, .)$, and hence the combiner can simply perform elementwise summation over its inputs, namely $Q_i(\mathbf{o}_t, \mathbf{h}_{t-1}, .) = Q_i(o_i^t, h_i^{t-1}, .) + \sum_{j \neq i} f_{enc}^{ij}(c_j^t)$. The combiner chooses the action with the $\epsilon$-greedy policy $\pi(.)$. Let $\theta_{local}^i$ and $\theta_{enc}^{ij}$ denote the set of parameters of the local action generators and the message encoders, respectively. To prevent the lazy agent problem [18] and decrease the model complexity, we make $\theta_{local}^i$ the same for all $i$, and make $\theta_{enc}^{ij}$ the same for all $i$ and $j$ ($j \neq i$). Accordingly, we can drop the corner scripts and use $\boldsymbol{\theta} = \{\theta_{local}, \theta_{enc}\}$ and $f_{enc}(.)$ to denote the agent network parameters and the message encoder.

## 4.2 Loss Function Definition

During the training phase, the message encoder and local action generator jointly learn to generate the best estimation on the action values. More specifically, we employ a mixing network (shown in Figure 1(b)) to aggregate the global action value functions $Q_i(\mathbf{o}_t, \mathbf{h}_{t-1}, a_i^t)$ from each agents $i$, and yields the joint action value function, $Q_{tot}(\mathbf{o}_t, \mathbf{h}_{t-1}, \mathbf{a}_t)$. To limit the variance of the messages from the other agents, we introduce an extra loss term on the variance of the outputs of the message encoders $f_{enc}(c_j^t)$. The loss function during the training phase is defined as:

$$L(\theta_{local}, \theta_{enc}) = \sum_{b=1}^{B} \sum_{t=1}^{T} \left[ (y_{tot}^b - Q_{tot}(\mathbf{o}_t^b, \mathbf{h}_{t-1}^b, \mathbf{a}_t^b; \boldsymbol{\theta}))^2 + \lambda \sum_{i=1}^{N} Var(f_{enc}(c_i^{t,b})) \right] \quad (1)$$

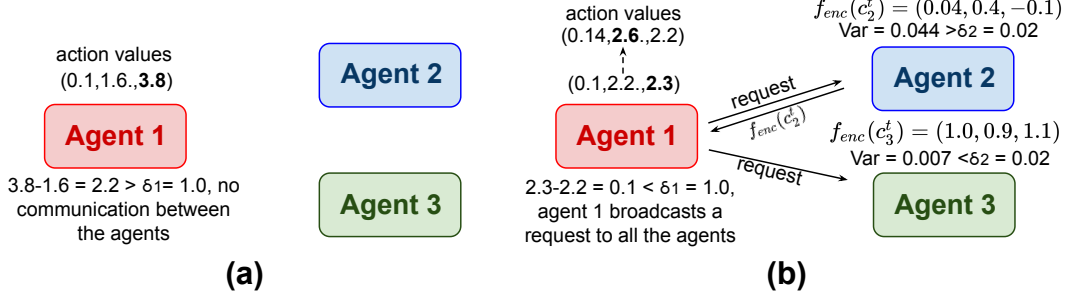

Figure 2: In (a), since the difference between the largest and second largest action values is greater than $\delta_1$, no communication is required. In (b), agent 1 broadcasts a request to agent 2 and 3, only agent 2 replies the request since the variance of $f_{enc}(c_2^t)$ is greater than $\delta_2$.

---

**Algorithm 1:** Communication protocol at agent $i$

1 **Input**: Confidence threshold of local actions $\delta_1$, threshold on variance of message encoder output $\delta_2$. Total number of agents N.
2 **for** $t \in T$ **do**
3      // **Decision on the action of itself**:
4      Compute local action values $Q_i(o_i^t, h_i^{t-1}, .)$. Denote $m_1, m_2$ the top two largest values of $Q_i(o_i^t, h_i^{t-1}, .)$.
5      **if** $m_1 - m_2 \geq \delta_1$ **then**
6          Let $Q_i(\mathbf{o}_t, \mathbf{h}_{t-1}, .) = Q_i(o_i^t, h_i^{t-1}, .)$.
7      **else**
8          Broadcast a request to the other agents, and receive the $f_{enc}(c_j^t)$ from $N_{reply}(N_{reply} \leq N)$ agents.
9          Let $Q_i(\mathbf{o}_t, \mathbf{h}_{t-1}, .) = Q_i(o_i^t, h_i^{t-1}, .) + \sum_{j=1}^{N_{reply}} f_{enc}(c_j^t)$.
10      // **Generating reply messages for the other agents**:
11      Calculate variance of $f_{enc}(c_i^t)$, if $Var(f_{enc}(c_i^t)) \geq \delta_2$, store $f_{enc}(c_i^t)$ in the buffer.
12      **if** $Var(f_{enc}(c_i^t)) \geq \delta_2$ **and** *Receive a request from agent j* **then**
13          Reply the request from agent j with $f_{enc}(c_i^t)$.

---

where $y_{tot}^b = r_t^b + \gamma max_{\mathbf{a}_{t+1}} Q_{tot}(\mathbf{o}_{t+1}^b, \mathbf{h}_t^b, \mathbf{a}_{t+1}; \boldsymbol{\theta}^-)$, $\boldsymbol{\theta}^-$ is the parameter of the target network which is copied from the $\boldsymbol{\theta}$ periodically, $Var(.)$ is the variance function and $\lambda$ is the weight of the loss on it. $b$ is the batch index. The replay buffer is refreshed periodically by running each agent network and selecting the action which maximizes $Q_i(\mathbf{o}_t, \mathbf{h}_{t-1}, .)$.

### 4.3 Communication Protocol Design

During the execution phase, at every timestep $t$, the agent $i$ first computes the local action value function $Q_i(o_i^t, h_i^{t-1}, .)$ and $f_{enc}(c_i^t)$. It then measures the confidence level on the local decision by computing the difference between the largest and the second largest element within the action values. An example is given in Figure 2(a). Assume agent 1 has three actions to select, and the output of the local action generator of agent 1 is $Q_1(o_1^t, h_1^{t-1}, .) = (0.1, 1.6, 3.8)$, and the difference between the largest and the second largest action values is $3.8 - 1.6 = 2.2$, which is greater than the threshold $\delta_1 = 1.0$. Given the fact that the variance of message encoder outputs $f_{enc}(c_j^t)$ from the agent 2 and 3 is relatively small due to the additional penalty term on variance in equation 1, it is highly possible that the global action value function $Q_1(\mathbf{o}_t, \mathbf{h}_{t-1}, .)$ also has the largest value in its third element. Therefore agent 1 does not have to talk to other agents to acquire $f_{enc}(c_j^t)$. Otherwise, agent 1 broadcasts a request to ask for help if its confidence level on the local decision is low. Because the request does not contain any actual data, it consumes very low bandwidth. Upon receiving the request, only the agents whose message has a large variance reply (Figure 2(b)), because their messages may change the current action decision of agent 1. This protocol not only reduces the communication overhead considerably, but also eliminates noisy, less informative messages that may impair the overall performance. The detailed protocol and operations performed at an agent $i$ is summarized in Algorithm 1.

# 5 Convergence Analysis

In this section, we analyze convergence of the learning process with the loss function defined in equation (1) under the tabular setting. For the sake of simplicity, we ignore the dependency of the action value function on the previous knowledge $\mathbf{h}_t$. To minimize equation (1), given the initial state $Q_0$, at iteration $k$, the $q$ values in the table is updated according to the following rule:

$$Q_{tot}^{k+1}(\mathbf{o}_t, \mathbf{a}_t) = Q_{tot}^k(\mathbf{o}_t, \mathbf{a}_t) + \eta_k \left[ r_t + \gamma max_{\mathbf{a}} Q_{tot}^k(\mathbf{o}_{t+1}, \mathbf{a}) - Q_{tot}^k(\mathbf{o}_t, \mathbf{a}_t) - \lambda \sum_{i=1}^{N} \frac{\partial Var(f_{enc}(c_i^t))}{\partial Q_{tot}^k(\mathbf{o}_t, \mathbf{a}_t)} \right] \quad (2)$$

where $\eta_k$, $Q_{tot}^k(.)$ are the learning rate and the joint action value function at iteration $k$ respectively. Let $Q_{tot}^*(.)$ denote the optimal joint action value function. We have the following result on the convergence of the learning process. A detailed proof is given in the supplementary materials.

**Theorem 1.** *Assume* $0 \leq \eta_k \leq 1$, $\sum_k \eta_k = \infty$, $\sum_k \eta_k^2 < \infty$. *Also assume the number of possible actions and states are finite. By performing equation 2 iteratively, we have* $||Q_{tot}^k(\boldsymbol{o}_t, \boldsymbol{a}_t) - Q_{tot}^*(\boldsymbol{o}_t, \boldsymbol{a}_t)|| \leq \lambda N G \ \forall \boldsymbol{o}_t, \boldsymbol{a}_t$, *as* $k \to \infty$, *where* $G$ *satisfies* $||\frac{\partial Var(f_{enc}(c_i^t))}{\partial Q_{tot}^k(\boldsymbol{o}_t, \boldsymbol{a}_t)}|| \leq G, \forall i, k, t, \boldsymbol{o}_t, \boldsymbol{a}_t$.

# 6 Experiment

We evaluated the performance of VBC on the StarCraft Multi-Agent Challenge (SMAC) [15]. StarCraft II [1] is a real-time strategy (RTS) game that has recently been utilized as a benchmark by the reinforcement learning community [14, 5, 13, 4]. In this work, we focus on the *decentralized micromanagement problem* in StarCraft II, which involves two armies, one controlled by the user (i.e. a group of agents), and the other controlled by the build-in StarCraft II AI. The goal of the user is to control its allied units to destroy all enemy units, while minimizing received damage on each unit. We consider six different battle settings. Three of them are *symmetrical battles*, where both the user and the enemy groups consist of 2 Stalkers and 3 Zealots (2s3z), 3 Stalkers and 5 Zealots (2s5z), and 1 Medivac, 2 Marauders and 7 Marines (MMM) respectively. The other three are *unsymmetrical battles*, where the user and enemy groups have different army unit compositions, including: 3 Stalkers for user versus 4 Zealots for enemy (3s_vs_4z), 6 Hydralisks for user versus 8 Zealots for enemy (6s_vs_8z), and 6 Zealot for user versus 24 Zerglings for enemy (6z_vs_24zerg). The unsymmetrical battles are considered to be harder than the symmetrical battles because of the difference in army size.

At each timestep, each agent controls a single unit to perform an action, including move[direction], attack[enemy_id], stop and no-op. Each agent has a limited *sight range* and *shooting range*, where shooting range is less than the sight range. The attack operation is available only when the enemies are within the shooting range. The joint reward received by the allied units equals to the total damage inflicted on enemy units. Additionally, the agents are rewarded 100 extra points after killing each enemy unit, and 200 extra points for killing the entire army. The user wins the battle only when the allied units kill all the enemies within the time limit. Otherwise the built-in AI wins. The input observation of each agent is a vector that consists of the following information of each allied unit and enemy unit in its sight range: relative $x$, $y$ coordinates, relative distance and agent type. For the detailed game settings, hyperparameters, and additional experiment evaluation over other test environments, please refer to supplementary materials.

## 6.1 Results

We compare VBC and several benchmark algorithms, including VDN [18], QMIX [14] and Sched-Net [8] for controlling allied units. We consider two types of VBCs by adopting the mixing networks of VDN and QMIX, denoted as VBC+VDN and VBC+QMIX. The mixing network of VDN simply computes the elementwise summation across all the inputs, and the mixing network of QMIX deploys a neural network whose weight is derived from the global state $\mathbf{s}_t$. The detailed architecture of this mixing network can be found in [14]. Additionally, we create an algorithm FC (full communication) by removing the penalty in Equation (1), and dropping the limit on variance during the execution phase (*i.e.*, $\delta_1 = \infty$ and $\delta_2 = -\infty$). The agents are trained with the same network architecture shown in Figure (1), and the mixing network of VDN is used. For SchedNet, at every timestep only $K$ out of $N$ agents can broadcast their messages by using *Top(k)* scheduling policy [8]. We usually set $K$ close to $0.5N$, that is, each time roughly half of the allied units can broadcast their messages. The

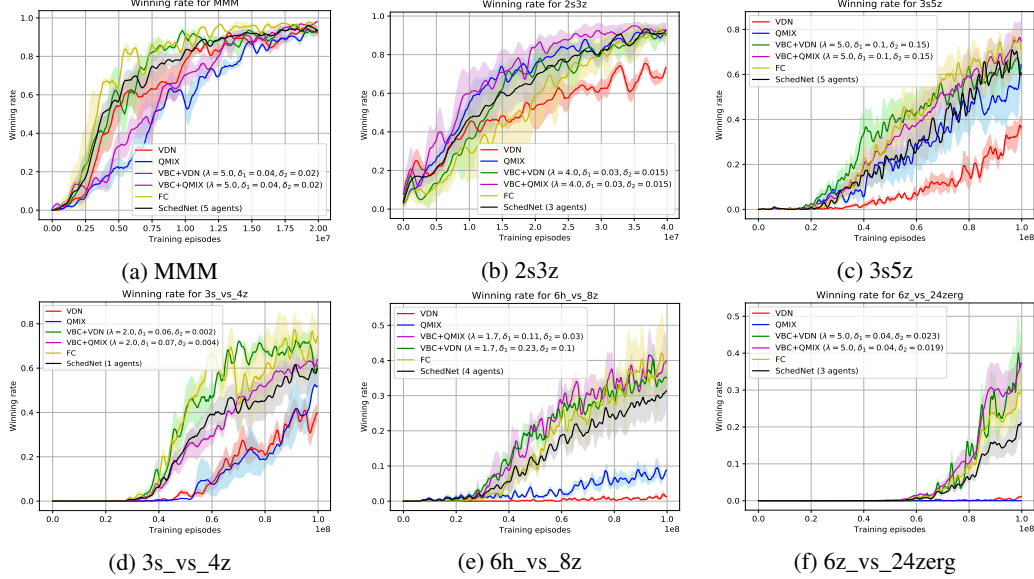

|  |  |  |
|:---:|:---:|:---:|
| (a) MMM | (b) 2s3z | (c) 3s5z |
| (d) 3s_vs_4z | (e) 6h_vs_8z | (f) 6z_vs_24zerg |

Figure 3: Winning rates for the six tasks, the shaded regions represent the 95% confidence intervals.

VBC are trained for different number of episodes based on the difficulties of the battles, which we describe in detail next.

To measure the convergence speed of each algorithm, we stop the training process and save the current model every 200 training episodes. We then run 20 test episodes and measure the winning rates for these 20 episodes. For VBC+VDN and VBC+QMIX, the winning rates are measured by running the communication protocol described in Algorithm 1. For easy tasks, namely MMM and 2s_vs_3z, we train the algorithms with 2 million and 4 million episodes respectively. For all the other tasks, we train the algorithms with 10 million episodes. Each algorithm is trained 15 times. Figure 3 shows the average winning rate and 95% confidence interval of each algorithm for all the six tasks. For hyperparameters used by VBC (*i.e.*, $\lambda$ used in equation (1), $\delta_1 and \delta_2$ in Algorithm 1), we first search for a coarse parameter range based on random trial, experience and message statistics. We then perform a random search within a smaller hyperparameter space. Best selections are shown in the legend of each figure.

We observe that the algorithms that involve communication (*i.e.*, SchedNet, FC, VBC) outperform the algorithms without communication (*i.e.*, VDN, QMIX) in all the six tasks. This is a clear indication that communication benefits the performance. Moreover, both VBC+VDN and VBC+QMIX achieve better winning rates than SchedNet, because SchedNet only allows a fixed number of agents to talk at every timestep, which prohibits some key information to exchange in a timely fashion. Finally, VBC achieves similar performance as FC and even outplays FC for some tasks (*e.g.*, 2s3z,6h_vs_8z, 6z_vs_24zerg). This is because a fair amount of communication between the agents are noisy and redundant. By eliminating these undesired messages, VBC is able to achieve both communication efficiency and performance gain.

## 6.2 Communication Overhead

We now evaluate the communication overhead of VBC. To quantify the amount of communication involved, we run Algorithm 1 and count the total number of pairs of agents $g_t$ that conduct communication for each timestep $t$, then divided by the total number of pairs of agents in the user group, $R$. In other words, the communication overhead $\beta = \sum_{t=1}^{T} g_t / RT$. An an example, for the task 3s_vs_4z, since the user controls 3 Stalkers, and the total number of agent pairs is $R = 3 \times 2 = 6$. Within these 6 pairs of agents, suppose that 2 pairs involve communication, then $g_t = 2$. Table 1 shows the $\beta$ of VBC+VDN, VBC+QMIX and SchedNet across all the test episodes at the end of the training phase of each battle. For SchedNet, $\beta$ simply equals the ratio between the number of allied agents that are allowed to talk and the total number of allied agents. As shown in Table 1, in contrast to ScheNet,

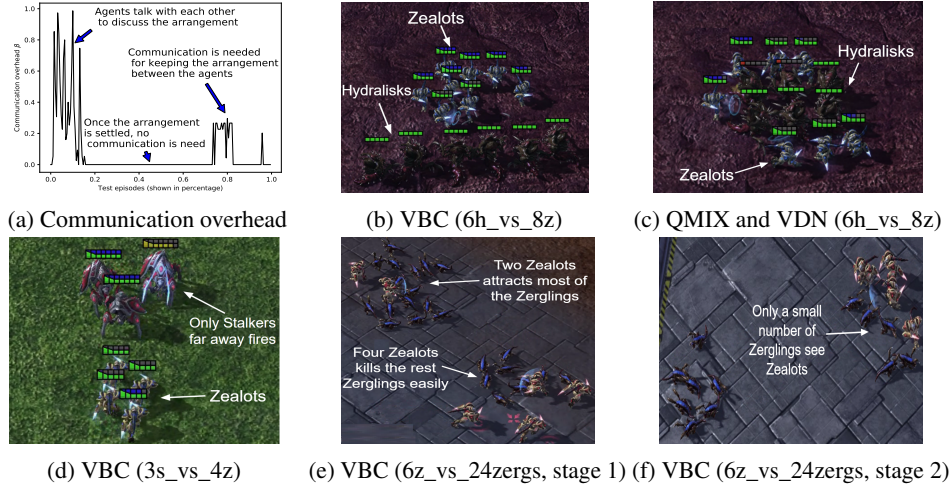

(a) Communication overhead      (b) VBC (6h_vs_8z)      (c) QMIX and VDN (6h_vs_8z)

(d) VBC (3s_vs_4z)      (e) VBC (6z_vs_24zergs, stage 1) (f) VBC (6z_vs_24zergs, stage 2)

Figure 4: Strategies and communication pattern for different scenarios

Table 1: Communication overhead

| $\beta$ | VBC+VDN | VBC+QMIX | SchedNet |
|---|---|---|---|
| **MMM** | 5.25% | 5.36% | 50% |
| **2s3z** | 4.33% | 4.68% | 60% |
| **3s5z** | 27.70% | 28.13% | 62.5% |
| **3s_vs_4z** | 5.07% | 5.19% | 33.3% |
| **6h_vs_8z** | 35.93% | 36.16% | 66.7% |
| **6z_vs_24zerg** | 12.13% | 13.35% | 50% |

VBC+VDN and VBC+QMIX produce $10\times$ lower communication overhead for MMM and 2s3z, and $2 - 6\times$ less traffic for the rest of tasks.

## 6.3 Learned Strategy

In this section, we examine the behaviors of the agents in order to better understand the strategies adopted by the different algorithms. We have made a video demo available at [2] for better illustration.

For unsymmetrical battles, the number of allied units is less than the enemy units, and therefore the agents are prone to be attacked by the enemies. This is exactly what happened for the QMIX and VDN agents on 6h_vs_8z, as shown in (Figure 4(c)). Figure 4(b) shows the strategy of VBC, all the Hydralisks are placed in a row at the bottom margin of the map. Due to the limited size of the map, the Zealots can not go beyond the margin to surround the Hydralisks. The Hydralisks then focus their fire to kill each Zealot. Figure 4(a) shows the change on $\beta$ for a sample test episode. We observe that most of the communication appears in the beginning of the episode. This is due to the fact that Hydralisks need to talk in order to arrange in a row formation. After the arrangement is formed, no communication is needed until the arrangement is broken due to the deaths of some Hydralisks, as indicated by the short spikes near the end of the episode. Finally, SchedNet and FC utilize a similar strategy as VBC. Nonetheless, due to the restriction on communication pattern, the row formed by the allied agents are usually not well formed, and can be easily broken by the enemies.

For 3s_vs_4z scenario, the Stalkers have a larger attack range than Zealots. All the algorithms adopt a kiting strategy where the Stalkers form a group and attack the Zealots while kiting them. For VBC and FC, at each timestep only the agents that are far from the enemies attack, and the rest of the agents (usually the healthier ones) are used as a shield to protect the firing agents (Figure 4(d)). Communication only occurs when the group are broken and need to realign. In contrast, VDN and QMIX do not have this attacking pattern, and all the Stalkers always fire simultaneously, therefore the Stalkers closest to the Zealots are get killed first. SchedNet and FC also adopt a similar policy as VBC, but the attacking pattern of the Stalkers is less regular, i.e., the Stalkers close to the Zealots also fire occasionally.

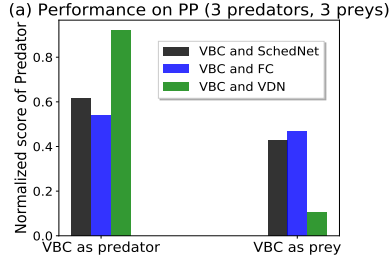

(a) Performance on PP (3 predators, 3 preys)

| (b) Results on Cooperative Navigation (#agents = 6) | | |
|---|---|---|
| Methods | Avg. dist | #collisions |
| VBC+VDN | 2.687 | 0.169 |
| SchedNet | 2.798 | 0.176 |
| FC | 2.990 | 0.161 |
| VDN | 3.886 | 1.872 |

Figure 5: (a) Results on PP with 3 predators and 3 prey. For SchedNet, we allow 1 predator/prey to broadcast messages. (b) Results of CN. For SchedNet, we allow 3 agents to broadcast messages.

6z_vs_24zerg is the toughest scenario in our experiment. For QMIX and VDN, the 6 Zealots are surrounded and killed by 24 Zerglings shortly after the episode starts. In contrast, VBC first separates the agents into two groups with two Zealots and four Zealots respectively (Figure 4(e)). The two Zealots attract most of the Zerglings to a place far away from the rest four Zealots, and are killed shortly. Due to the limit sight range of the Zerglings, they can not find the rest four Zealots. On the other side, the four Zealots kill the small part of Zerglings easily and search for the rest Zerglings. The four Zealots take advantage of the short sight of the Zerglings. Each time the four Zealots adjust their positions in a way such that they can only be seen by a small number of the Zerglings, the baited Zerglings are then killed easily (Figure 4(f)). For VBC, the communication only occurs in the beginning of the episode when the Zealots are separated into two groups, and near the end of the episode when four Zealots adjust their positions. Both FC and SchedNet learn the strategy of splitting the Zealots into two groups, but they fail to fine-tune their positions to kill the remaining Zerglings.

For symmetrical battles, the tasks are less challenging, and we see less disparities on performances of the algorithms. For 2s3z and 3s5z, the VDN agents attack the enemies blindly without any cooperation. The QMIX agents learn to focus firing and protect the Stalkers. The agents of VBC, FC and SchedNet adopt a more aggressive policy, where the allied Zealots try to surround and kill the enemy Zealots first, and then attack the enemy Stalkers by collaborating with the allied Stalkers. This is extremely effective because Zealots counter Stalkers, so it is important to kill the enemy Zealots before they damage allied Stalkers. For VBC, the communication occurs mostly when the allied Zealots try to surround the enemy Zealots. For MMM, almost all the methods learn the optimal policy, namely killing the Medivac first, then attack the rest of the enemy units cooperatively.

### 6.4 Evaluation on Cooperative Navigation and Predator-prey

To demonstrate the applicability of VBC in more general settings, we have tested VBC for two more scenarios: (1) Cooperative Navigation (CN) which is a cooperative scenario, and (2) Predator-prey (PP) which is a competitive scenario. The game settings are the same as what are used in [10] and [8], respectively. We train each method until convergence and test the result models for 2000 episodes. For PP, we make the agents of VBC compete against the agents of other methods, and report the normalized score of Predator (Figure 5(a)). For CN we report the average distance between agents and their destinations, and average number of collisions (Figure 5(b)). We notice that methods which allow communication (*i.e.*, SchedNet, FC, VBC) outperform the others for both tasks, and VBC achieves the best performance. Moreover, in both scenarios, VBC incurs $10\times$ and $3\times$ lower communication overhead than FC and SchedNet respectively. In CN, most of the communication of VBC occurs when the agents are close to each other to prevent collisions. In PP, the communication of VBC occurs mainly to rearrange agent positions for better coordination. These observations confirm that VBC's can be applied to a variety of MARL scenarios with great effectiveness.

## 7 Conclusion

In this work, we propose VBC, a simple and effective approach to achieve efficient communication among agents in MARL. By constraining the variance of the exchanged messages during the training phase, VBC improves communication efficiency while enabling better cooperation among the agents. The test results of multiple MARL benchmarks indicate that VBC outperforms the other state-of-the-art methods significantly in terms of both performance and communication overhead.

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
