[Supplementary Material]

# Supplementary Materials for Efficient Communication in Multi-Agent Reinforcement Learning via Variance Based Control

## 1 Proof for theorem 1

**Theorem 1.** *Assume $0 \leq \eta_k \leq 1$, $\sum_k \eta_k = \infty$, $\sum_k \eta_k^2 < \infty$. Also assume the number of possible actions and states are finite. By performing equation 2 iteratively, we have $||Q_{tot}^k(\boldsymbol{o}_t, \boldsymbol{a}_t) - Q_{tot}^*(\boldsymbol{o}_t, \boldsymbol{a}_t)|| \leq \lambda NG \; \forall \boldsymbol{o}_t, \boldsymbol{a}_t$, as $k \to \infty$, where $G$ satisfies $||\frac{\partial Std(f_{enc}(c_i^t))}{\partial Q_{tot}^k(\boldsymbol{o}_t, \boldsymbol{a}_t)}|| \leq G, \forall i, k, t, \boldsymbol{o}_t, \boldsymbol{a}_t$.*

*Proof.* The proof is based on [4] and [5]. If we subtract $Q_{tot}^*(\mathbf{o}_t, \mathbf{a}_t)$ from equation 2 in the paper, and rearrange the equation, we get:

$$\delta^{k+1}(\mathbf{o}_t, \mathbf{a}_t) = (1-\eta_k)\delta^k(\mathbf{o}_t, \mathbf{a}_t) + \eta_k \left[ r_t + \gamma max_{\mathbf{a}} Q_{tot}^k(\mathbf{o}_{t+1}, \mathbf{a}) - Q_{tot}^*(\mathbf{o}_t, \mathbf{a}_t) - \lambda \sum_{i=1}^{N} \frac{\partial Std(f_{enc}(c_i^t))}{\partial Q_{tot}^k(\mathbf{o}_t, \mathbf{a}_t)} \right] \tag{1}$$

where $\delta^k(\mathbf{o}_t, \mathbf{a}_t) = Q_{tot}^k(\mathbf{o}_t, \mathbf{a}_t) - Q_{tot}^*(\mathbf{o}_t, \mathbf{a}_t)$. Let $r_t + \gamma max_{\mathbf{a}} Q_{tot}^k(\mathbf{o}_{t+1}, \mathbf{a}) - Q_{tot}^*(\mathbf{o}_t, \mathbf{a}_t) = F_k(\mathbf{o}_t, \mathbf{a}_t)$, and $\sum_{i=1}^N \frac{\partial Std(f_{enc}(c_i^t))}{\partial Q_{tot}^k(\mathbf{o}_t, \mathbf{a}_t)} = U_k(\mathbf{o}_t, \mathbf{a}_t)$ we have:

$$\delta^{k+1}(\mathbf{o}_t, \mathbf{a}_t) = (1 - \eta_k)\delta^t(\mathbf{o}_t, \mathbf{a}_t) + \eta_k \left[ F_k(\mathbf{o}_t, \mathbf{a}_t) - \lambda U_k(\mathbf{o}_t, \mathbf{a}_t) \right] \tag{2}$$

Decompose $\delta^{k+1}(\mathbf{o}_t, \mathbf{a}_t)$ into two random processes, $\delta_1^{k+1}(\mathbf{o}_t, \mathbf{a}_t)$ and $\delta_2^{k+1}(\mathbf{o}_t, \mathbf{a}_t)$, where $\delta^{k+1}(\mathbf{o}_t, \mathbf{a}_t) = \delta_1^{k+1}(\mathbf{o}_t, \mathbf{a}_t) + \delta_2^{k+1}(\mathbf{o}_t, \mathbf{a}_t)$, we have the following two random iterative processes:

$$\delta_1^{k+1}(\mathbf{o}_t, \mathbf{a}_t) = (1 - \eta_k)\delta_1^k(\mathbf{o}_t, \mathbf{a}_t) + \eta_k F_k(\mathbf{o}_t, \mathbf{a}_t) \tag{3}$$

$$\delta_2^{k+1}(\mathbf{o}_t, \mathbf{a}_t) = (1 - \eta_k)\delta_2^k(\mathbf{o}_t, \mathbf{a}_t) - \eta_k \lambda U_k(\mathbf{o}_t, \mathbf{a}_t) \tag{4}$$

From Theorem 1 of [5], we know that $\delta_1^{k+1}(\mathbf{o}_t, \mathbf{a}_t)$ converges to zero w.p. 1. From equation 6, we notice that:

$$||\delta_2^{k+1}(\mathbf{o}_t, \mathbf{a}_t)|| \leq ||(1 - \eta_k)\delta_2^k(\mathbf{o}_t, \mathbf{a}_t)|| + \eta_k \lambda ||U_t(\mathbf{o}_t, \mathbf{a}_t)|| \tag{5}$$

$$\leq (1 - \eta_k)||\delta_2^k(\mathbf{o}_t, \mathbf{a}_t)|| + \eta_k \lambda NG \tag{6}$$

Therefore we have $||\delta_2^{k+1}(\mathbf{o}_t, \mathbf{a}_t)|| - \lambda NG \leq (1 - \eta_k)(||\delta_2^k(\mathbf{o}_t, \mathbf{a}_t)|| - \lambda NG)$. This is linear in $\delta_2^{k+1}(\mathbf{o}_t, \mathbf{a}_t)$ and $||\delta_2^{k+1}(\mathbf{o}_t, \mathbf{a}_t)|| - \lambda NG$ will converge to a nonpositive number as k approaches infinity. However, because $||\delta_2^{k+1}(\mathbf{o}_t, \mathbf{a}_t)||$ is always greater or equal to zero, hence $||\delta_2^{k+1}(\mathbf{o}_t, \mathbf{a}_t)||$ must converge to a number between 0 and $\lambda NG$. Therefore we get:

$$||Q_{tot}^k(\mathbf{o}_t, \mathbf{a}_t) - Q_{tot}^*(\mathbf{o}_t, \mathbf{a}_t)|| = ||\delta^k(\mathbf{o}_t, \mathbf{a}_t)|| \tag{7}$$

$$= ||\delta_1^k(\mathbf{o}_t, \mathbf{a}_t) + \delta_2^k(\mathbf{o}_t, \mathbf{a}_t)|| \tag{8}$$

$$\leq ||\delta_1^k(\mathbf{o}_t, \mathbf{a}_t)|| + ||\delta_2^k(\mathbf{o}_t, \mathbf{a}_t)|| \tag{9}$$

$$\leq \lambda NG \tag{10}$$

as k approaches infinity. $\square$

Table 1: Agent types of the six battles

| Symm. | 2s3z | MMM | 3s5z |
|---|---|---|---|
| User | 2 Stalkers & 3 Zealots | 1 Medivac, 2 Marauders & 7 Marines | 3 Stalkers & 5 Zealots |
| Enemy | 2 Stalkers & 3 Zealots | 1 Medivac, 2 Marauders & 7 Marines | 3 Stalkers & 5 Zealots |
| **Unsymm.** | **3s_vs_4z** | **6h_vs_8z** | **6z_vs_24zerg** |
| User | 3 Stalkers | 6 Hydralisks | 6 Zealots |
| Enemy | 4 Zealots | 8 Zealots | 24 Zerglings |

# 2 Experiment settings and hyperparameters

In this section, we describe in detail the experiment settings.

## 2.1 StarCraft micromanagement challenges

StarCraft II [1] is a real-time strategy (RTS) game that has recently been utilized as a challenging benchmark by the reinforcement learning community [7, 3, 6, 2]. In this work, we concentrate on the *decentralized micromanagement problem* in StarCraft II. Specifically, the user controls an army that consists of several army units (agents). We consider the battle scenario where two armies, one controlled by the user, and the other controlled by the build-in StarCraft II AI, are placed on the same map and try to defeat each other. The agent type can be different between the two armies, and the agent type can also be different within the same army. The goal of the user is to control the allied units wisely to kill all the enemy units, while minimizing the damage on the health of each individual agent. The difficulty of the computer AI is set to *Medium*. We consider six different battle settings, which is shown in Table 1. Among these six settings, three are *symmetrical battles*, where the user group and the enemy group are identical in terms of type and quantity of agents. The other three are *unsymmetrical battles*, where the agent type of user groups and enemy group are different, and user group contains less number of allied units than the enemy group. The unsymmetrical battles is consider to be harder than the symmetrical battles because of the difference in the army size.

During execution, each agent are allowed to perform the following actions: move[direction], attack[enemy_id], stop and no-op. There are four directions for the 'move' operation: east, west, south, or north. Each agent can only attack the enemy within its *shooting range*. The Medivacs can only heal the partner by performing the heal[partner_id] instead of attacking the enemies. The number of possible actions for an agent ranges from 11 (2s3z) to 30 (6z_vs_24zerg). Each agent has a limited *sight range*, and can only receive information from the partners or enemies within the sight range. Furthermore, the shooting range is smaller than the sight range so the agent can not attack the opponents without observing them, and the attack operation is not available when the opponents are outside the shooting range. Finally, agents can only observe the live agents within the sight range, and can not discern the agents that are dead or outside the range. At each timestep, the joint reward received by the allied units equals to the total damage on the health levels of the enemy units. Additionally, the agents are rewarded 100 extra points after killing each enemy unit, and 200 extra points for killing all the enemies. The user group wins the battle only when the allied units kill all the enemies within the time limit. The user group loses the battle if either all the allied units are killed, or the time limit reaches. The time limit for different battles are: 120 timesteps for 2s3z and MMM, 150 for 3s5z and 3s_vs_4z, and 200 for 6h_vs_8z and 6z_vs_24zerg.

The input observation of each agent consisting of a vector which involves the following information of each allied unit and enemy unit in its sight range: relative x,y coordinates, relative distance and agent type. For the mixing network of QMIX, the global state vector $\mathbf{s}_t$ contains the following elements:

1. Shield levels, health levels and cooldown levels of all the units at $t$.

2. The actions taken by all the units at $t - 1$.

3. The x,y coordinates of all the units relative to the center of the map at $t$.

For all the six battles, each allied or enemy agent has a sight range of 9 and shooting range of 6 for all types of agents. For additional information, please refer to [8].

## 2.2 Hyperparameter

For network of agent $i$, at timestep $t$, raw observation $o_i^t$ is first passed through a single-layer MLP, which outputs a intermediate result with size of 64. The GRU then takes this intermediate result, as well as the hidden states $h_i^{t-1}$ from the previous timestep and generates $h_i^t$ and $c_i^t$. Both $h_i^t$ and $c_i^t$ has a size of 64. The $c_i^t$ is then passed through a FC layer, which generates the local action-value function $Q_i(o_i^t, h_i^t, .)$. The message encoder contains two FC layers with 196 and 64 units respectively. The combiner performs elementwise summation on the outputs of local action generator and the message encoders.

During the training, we set the $\gamma = 0.99$ and decrease $\epsilon$ linearly from 1.0 to 0.05 over the first 200000 timesteps and keep it to 0.05 for the rest of the learning process. The replay buffers stores the most recent 5000 episode. We perform a test run for every 200 training episodes to update the replay buffer. The training batch size is set to 32 and the test batch size is set to 8. We adopt a RMSprop optimizer with a learning rate $\eta = 5 \times 10^{-4}$ and $\alpha = 0.99$.