[Reviews · NeurIPS 2019]

Reviewer 1



1- The messages exchanged are a bunch of floats. What these messages mean or represent, and how agents can make sense of them looks currently as black magic. There's a lack of explainability, and intuition of "but somehow it works, so..." with no analysis of the messages meaning. 2- The way the "confidence level" of an agent is computed is somewhat naive. 3- Equation 1 is unclear: which variance exactly is computed? The variance between agents' messages or the variance of the messages coming from one agent? 4- The environment chosen (StarCraft) seems adequate, and results are convincing. 5- I found the interpretations offered by the authors of the strategies developed by the agents while communicating interesting and welcome.

Reviewer 2



The paper contributes to the overall class of MARL algorithms as another simple communication method that improves performance with reduced communication costs. - I am a bit worried about the methods narrow application. It was only evaluated on a collection of similar Starcraft II environments. It also only works on cooperative environments. - Line 111 the Q function targets should be optimized over s_{t+1}, not s_{t}. I think this is just a typo and does not reflect in the results. - I do find it odd that MADDPG (Multi-Agent Actor-Critic for Mixed Cooperative-Competitive Environments) was not referenced in this paper. It is very related and has a form of implicit communication. - The change to the learning loss is simple. - There is little discussion on the learning hyper parameter introduced and the messaging thresholds. How are these chosen? How sensitive is the method to these values? It is not explicitly said what values are used for the experiments. I assume the same from the figures. After going over the author response I appreciate the extra analysis put into comparing the method to MADDPG to make sure it is state of the art. It is good to compare these methods across previous benchmarks to show improvement. While the additional hyperparameter analysis is helpful it is a bit obvious of what is normally done. Some discussion on the effects of specific settings might shed more light on how the method works. I have updated my scoring.

Reviewer 3



The paper is well written and easy to read. I very much enjoyed reading the paper. 1. Line 151: an individual agent can access the global observation and global history only through the conditioned messages. Is that right? If so, please make it explicit for better clarity. 2. Line 154: The fact that the combiner is just doing element wise addition can also be motivated as each agent trying to pass the message which could be the value of each action from that agent’s point of view. This could also motivate the variance based control loss because when there is not much variance in the message, then that agent do not have any preference over which action to choose and hence its message can be safely ignored. 3. It is not clear whether the communication protocol is used during the training or only during the testing time. I assume that you are using the same communication protocol even during training. Please explain this. I did not verify the correctness of the proof. ########################################################## Originality: The paper proposes a novel variance based loss to reduce communication overhead in a MARL setting. Quality: The work is good enough to be accepted at NeurIPS. Clarity: The paper is well written. I have given few comments above to improve the clarity in the presentation. Significance: Definitely a significant contribution to MARL. ###################################################### Minor comments: 1. Line 232: derivation -> deviation.

[Author Response · NeurIPS 2019]

We thank the reviewers for their insightful comments. We summarize the questions from each reviewer below and address them separately. We will incorporate the feedback and suggestions into the next revision of the paper.

**(Reviewer 1) Q1: Clarify what information is exchanged between the agents.**
A: The messages exchanged between the agents generally convey agent status information (location, health status, etc.) that are crucial for executing group strategy. Taking 6h_vs_8z as an example, we measured run-time agent communication pattern during formation stage (Figure 1(c)). Initially (step 80), the agents require frequent communication to build the formation. Overtime, communication level gradually decreases as agents move to the right position (step 250,430). As agent conditions are almost unchanged except for their locations during formation, it can be inferred that messages contain location information. We can also design similar experiments to infer the meaning of other types of messages.

**(Reviewer 1) Q2: Which real life applications would benefit most from the proposed approach?**
A: VBC is most beneficial to multi-agent systems that require quick decision making and low communication overhead. An example application is *Real-Time Traffic Signal Control*, where hundreds of thousands of traffic signal devices need to make cooperative decisions within tens of seconds. VBC can drastically reduce the communication cost and decrease latency on decision-making, as quite a number of decisions can be made locally. *Swarm Robotics* that are used in rescue missions and remote sensing services can also benefit from the bandwidth reduction and improve responsiveness offered by VBC in dynamic settings. VBC can also be deployed on *autonomous driving*, *cellular base station control*, etc.

**(Reviewer 1) Q3: Motivate the choice for calculating the confidence. What if two actions are quasi-optimal.**
A: We have tried other methods for calculating the confidence. For example, the variance of the local action vector is a natural choice for measuring the confidence. However, we found variance is not a good criteria in many cases. For example, when local action vector is $(10, 0.5, 0, 0.5)$, its variance is relatively high, but the action can be decided confidently (first action). When the action vector is $(10, 0, 10, 0)$, the variance is high but the best action is undecidable. In both cases, our approach, which is to compute the difference between the $1st$ and $2nd$ best local action scores provides a more direct measure of the action confidence. If the difference is small (i.e. two actions are quasi-optimal), the agent will request additional information from other agents, and then select the action with the highest action value.

**(Reviewer 1) Q4: Equation (1) is unclear: which variance is computed?**
Equation (1) minimizes the square loss as well as the sum of variance of the message from each agent.

**(Reviewer 2) Q1: Evaluate on a few other environments and compare with MADDPG.**
A: We evaluated MADDPG and found VBC achieves higher winning rates than MADDPG for all the six scenarios. Due to space limit, we only report the results for two StarCraft scenarios (Figure 1(a,b)). Furthermore, we evaluate the algorithms for two more scenarios: (1) Cooperative Navigation (CN) which is a cooperative scenario, and (2) Predator-prey (PP) which is a competitive scenario. The game settings are the same as in MADDPG paper. We train each method until convergence and test the result models for 2000 episodes. For PP, we make the agents of VBC to compete against the agents of other methods, and report the normalized score of VBC (Figure 1(d)). For CN we report the average distance between agents and their destinations, and average number of collisions (Figure 1(e)). We notice that methods which allows communication (*i.e.*, SchedNet, FComm, VBC) outperform the others (*i.e.*, VDN, MADDPG) for both tasks, and VBC achieves the best performance. Moreover, VBC incurs a communication overhead of $10.07\%$ and $8.80\%$ for PP and CN respectively, which is lower than SchedNet ($33\%$ and $50\%$). In CN, most of the communication of VBC occurs when the agents are close to each other to prevent collisions. In PP, the communication of VBC occurs mainly to rearrange agent positions for better coordination. We will cite MADDPG in our paper later.

**(Reviewer 2) Q2: I would like more discussion on the methods and introduced hyperparameters.**
A: For hyperparameters used by VBC (*i.e.*, regularization constant of variance, thresholds on confidence and message variance), we first search for a coarse parameter range based on random trial, experience and message statistics. We then perform a random search within a smaller hyperparameter space. Best selections are shown in Figure 3 of the paper.

**(Reviewer 4) Q1:An agent can access the global observation and history only through messages. Correct?**
A: Correct, we will make it clear in the next revision of the paper.

**(Reviewer 4) Q2: I assume that you are using the same communication protocol during training.**
A: That's correct, we will make it explicit in the next version of the paper.

Figure 1: (a) and (b) show the performance of MADDPG on two StarCraft scenarios. In (c) thin/bold edges represent uni/bidirectional communication respectively. (d) Results on PP with 3 predators and 3 prey. (e) shows results of CN.

[Meta-Review · NeurIPS 2019]

The paper proposes Variance Based Control (VBC) of communications in cooperative multi-agent RL settings. This results in agents sending only high-variance messages. As noted in the Abstract, VBC achieved 2x-10x reduction in communication overhead compared to state-of-the-art MARL settings. The paper also gives a proof of convergence in a tabular setting. In the initial reviews, R4 gave strongest support with a score of 9, while R1 and R2 gave positive overall scores but only at marginally above threshold (6). After receiving the author feedback, there were minimal updates to the original reviews, e.g., R2 said "After going over the author response I appreciate the extra analysis put into comparing the method to MADDPG to make sure it is state of the art. It is good to compare these methods across previous benchmarks to show improvement. While the additional hyperparameter analysis is helpful it is a bit obvious of what is normally done. Some discussion on the effects of specific settings might shed more light on how the method works." There was not a lot of discussion between individual reviewers. It seemed that everyone was satisfied with their individual scoring based on the positive or negative notes mentioned in the individual reviews. I think this explains the discrepancy of scoring by R4 vs. R1 and R2. R4's review focused focused on major issues of Originality, Quality, Clarity, and Significance, and I agree with R4 that these aspects are quite strong in the paper. On the other hand, R1 and R2 gave additional focus to lower-level issues and questions, for example: -- "Lack of explainability of exchanging a bunch of floats." -- "Confidence level is naive" -- "Unsure of significance" -- "concerned about narrow application" -- "how to set hyper-params"